# Loss of SIRT1 inhibits hematopoietic stem cell aging and age-dependent mixed phenotype acute leukemia

Zhiqiang Wang [1,2], Chunxiao Zhang [1], Charles David Warden [3], Zheng Liu[4], Yate-Ching Yuan[4], Chao Guo[4], Charles Wang [4,8], Jinhui Wang[3], Xiwei Wu[3], Richard Ermel[5], Steven L. Vonderfecht[6], Xiuli Wang[2], Christine Brown[2], Stephen Forman [2], Yaling Yang[7], M. James You[7] & WenYong Chen [1✉]

Aging of hematopoietic stem cells (HSCs) is linked to various blood disorders and malignancies. SIRT1 has been implicated in healthy aging, but its role in HSC aging is poorly understood. Surprisingly, we found that *Sirt1* knockout improved the maintenance of quiescence of aging HSCs and their functionality as well as mouse survival in serial bone marrow transplantation (BMT) recipients. The majority of secondary and tertiary BMT recipients of aging wild type donor cells developed B/myeloid mixed phenotype acute leukemia (MPAL), which was markedly inhibited by *Sirt1* knockout. SIRT1 inhibition also reduced the growth and survival of human B/myeloid MPAL cells. *Sirt1* knockout suppressed global gene activation in old HSCs, prominently the genes regulating protein synthesis and oxidative metabolism, which may involve multiple downstream transcriptional factors. Our results demonstrate an unexpected role of SIRT1 in promoting HSC aging and age-dependent MPAL and suggest SIRT1 may be a new therapeutic target for modulating functions of aging HSCs and treatment of MPAL.

[1] Department of Cancer Biology, Beckman Research Institute, City of Hope, Duarte, CA 91010, USA. [2] Department of Hematology and Hematopoietic Cell Transplantation, City of Hope National Medical Center, Duarte, CA 91010, USA. [3] Integrative Genomics Core, Department of Molecular and Cellular Biology, Beckman Research Institute, City of Hope, Duarte, CA 91010, USA. [4] Department of Molecular Medicine, Beckman Research Institute, City of Hope, Duarte, CA 91010, USA. [5] Center for Comparative Medicine, Beckman Research Institute, City of Hope, Duarte, CA 91010, USA. [6] Consultant in Veterinary Pathology, Reno, NV 89521, USA. [7] Department of Hematopathology, The University of Texas MD Anderson Cancer Center, Houston, TX 77030, USA. [8]Present address: Department of Basic Sciences, Loma Linda University School of Medicine, Loma Linda, CA 92350, USA. ✉email: wechen@coh.org

Aging population has increased substantially along with the prolonged lifespan in the world. As a result, a spectrum of age-dependent diseases including a variety of hematological disorders and malignancies become prevalent[1–3]. Hematopoietic stem cells (HSCs) play a key role in the homeostasis of the hematopoietic hierarchical system[4–6]. Aging of HSCs has been linked to multiple hematological dysfunctions and pathological changes including myeloid skewing, lymphoid deficiency, diminished immune competence, anemia, oligoclonal hematopoiesis, myelodysplastic syndrome, and hematological malignancies[6–9]. Among them, mixed phenotype acute leukemia (MPAL) is a rare heterogeneous group of high risk leukemia with features of both acute myeloid leukemia (AML) and acute lymphoblastic leukemia (ALL) because of co-expression of antigens of myeloid and lymphoid lineages on leukemia cells or co-presence of immunophenotypically distinct leukemia populations[10]. MPAL is classified as B/Myeloid (B/M), T/myeloid (T/M), MLL rearranged, BCR-ABL positive, and MPAL not otherwise specified[11]. MPAL is treated by either an AML or ALL protocol with ALL regimen producing better survival outcome. Notably, MPAL has much poorer prognosis in adults with median survival of only 11 months compared to 139 months in children[12], indicating a negative impact of aging on the outcome of MPAL. Integrative genomic analyses of MPAL have revealed numerous genetic and epigenetic alterations[13–17]. It is shown that the founding lesions of MPAL arise in primitive hematopoietic stem/progenitor cells, and that these lesions and the cell of origin likely prime MPAL cells for lineage promiscuity[15]. However, the mechanisms for leukemogenesis of MPAL and how hematopoietic stem/progenitor cell aging may impact MPAL progression remain unclear in part due to a lack of proper mouse models for MPAL. Currently, targeted therapy for MPAL is not available.

SIRT1 is a nicotinamide adenine dinucleotide-dependent protein deacetylase. SIRT1 is involved in diverse physiological and pathological processes including cell stress resistance, metabolism, DNA damage repair, and differentiation[18], and may promote healthy aging in mice[19–21]. However, SIRT1 expression is increased in various hematological malignancies, including chronic myeloid leukemia (CML)[22,23], some subtypes of AML[24], B-ALL[25], and T cell leukemia and lymphoma[26,27]. SIRT1 is upregulated in leukemia stem/progenitor cells in CML and some AML, and plays pivotal roles in leukemogenesis, acquiring mutations and drug resistance[22–24,28–30]. Sirt1 deficiency in mice causes early developmental defects[31,32]. But in adult Sirt1-null mice that survive developmental defects, they exhibit nearly normal HSC functions, hematopoiesis and lineage differentiation[22,33,34]. However, the role of SIRT1 in HSC aging is poorly understood and it is unknown if SIRT1 may play a role in MPAL.

In the current study, we examined the impact of Sirt1 knockout on HSC aging by serial bone marrow transplantation (BMT) in BALB/c mice. Surprisingly, we found that Sirt1 loss improved aging HSC functions in the knockout mice and primary BMT recipients. When the cumulative age of wild type donor bone marrow reached >31 months in serial BMT recipients, mice developed a high frequency of B/myeloid MPAL. Remarkably, Sirt1 knockout substantially inhibited B-cell lineage differentiation block and MPAL development. We found that SIRT1 protein was over-expressed in primary human B/myeloid MPAL, and SIRT1 knockout or inhibition suppressed human MPAL cell growth and survival. We further identified key pathways and genes that were upregulated in aging wild type HSCs but suppressed by Sirt1 knockout, which may contribute to MPAL development. We discussed the implication of our findings for aging research.

## Results

### Sirt1 knockout inhibited HSC aging in BALB/c mice.

We used a consecutive fostering protocol to generate constitutive Sirt1 knockout (KO) mice in BALB/c background[35], a strain with a median lifespan of 23–25 months. We previously showed that long-term HSCs in BALB/c mice exclusively reside in lineage-negative side population (SP hereafter)[35]. To examine the effect of Sirt1 loss on HSC aging, we compared old (16–20 months) Sirt1$^{-/-}$ (OKO) vs. Sirt1$^{+/+}$ (OWT) littermates, with young (2–3 months) wild type (YWT) BALB/c mice as references. We only used YWT for comparison because of a lack of significant difference in hematopoiesis of young Sirt1$^{-/-}$ and Sirt1$^{+/+}$ mice[22,35]. All Sirt1$^{-/-}$ mice in the BALB/c strain suffered some early development-related health issues including lower body-weight and a microphthalmia-like condition for the entire life. About half of Sirt1$^{-/-}$ mice lived through adulthood in this strain. We found that both OWT and OKO mice had higher neutrophil counts in blood and myeloid cell percentage in bone marrow than YWT, accompanied with lower bone marrow B cell but higher T cell percentage than YWT (Fig. 1a, b). These changes were more prominent in OKO mice, possibly related to the microphthalmia-like condition of OKO mice that increased ocular discharge causing facial inflammation. OWT, but not OKO, mice had reduced red blood cell counts (Fig. 1a). OWT mice expanded the SP HSC pool by about 6 folds over YWT mice (Fig. 1c), similar to that seen in old C57BL/6 mice[9], which more significantly impacted the lower part of SP (Fig. 1d). Surprisingly, Sirt1 knockout inhibited this age-dependent SP expansion (Fig. 1c, d). Consistently, quiescent cells (G0) within SP were significantly lower in OWT than YWT, but this change was reversed in OKO (Fig. 1e). The reduced cell cycle entry in OKO SP cells is in contrast to a slight increase of cell cycle entry in young Sirt1$^{-/-}$ SP[35], suggesting distinct functions of Sirt1 in aging HSCs. However, Sirt1 mRNA did not change significantly in OWT vs. YWT SP (Supplementary Fig. 1). We found that OKO mouse bone marrow cells had significantly higher yields of methylcellulose colony-forming units (CFUs) for GEMM, GM, CFU-E and BFU-E than OWT (Fig. 1F), indicating that OKO HSCs were functionally better than OWT HSCs. In line with Sirt1 KO, CFU-GEMM and CFU-GM colonies were significantly increased in old BALB/c and C57BL/6 mouse bone marrow cells treated with a SIRT1 inhibitor Ex-527 or nicotinamide (Supplementary Fig. 2).

To examine if the effect of Sirt1 KO is HSC cell autonomous, we performed serial BMT. Syngeneic BMT in BALB/c mice was performed with old male donor cells to YWT female recipients (unless specified otherwise), and was tracked by SRY analysis and Sirt1 KO genotyping as described[35]. We used the W–X–Y–Z serial number to code for BMT series. W, X, Y and Z denote the ages (months) of original donors, 1st round BMT recipients, 2nd round recipients, and 3rd round recipients, respectively. In the 1st round BMT 14-8 series (14-month-old donors, and 8-month-old recipients analyzed), mice receiving wild type bone marrow cells exhibited significantly lower WBC, lymphocyte and red blood cell counts in blood than the recipients of Sirt1$^{-/-}$ cells (Fig. 2a). When the cumulative age of bone marrow got older as in the 20-10 series, the further skewed lineage differentiation with myeloid expansion and reduced B cells was seen in some recipients of OWT cells (Fig. 2b). Similar to aging donor mice, SP was expanded more in the recipients of OWT cells than of OKO cells (Fig. 2c). In addition, CD150$^+$ SP cells were significantly increased to over 80% in the recipients of OWT cells (Fig. 2d). These results are consistent with a report that CD150$^+$ SP cells are increased during aging to favor myeloid differentiation[36], and show that Sirt1 loss inhibits skewed lineage differentiation. Although the lifespan of the 1st round BMT recipients of

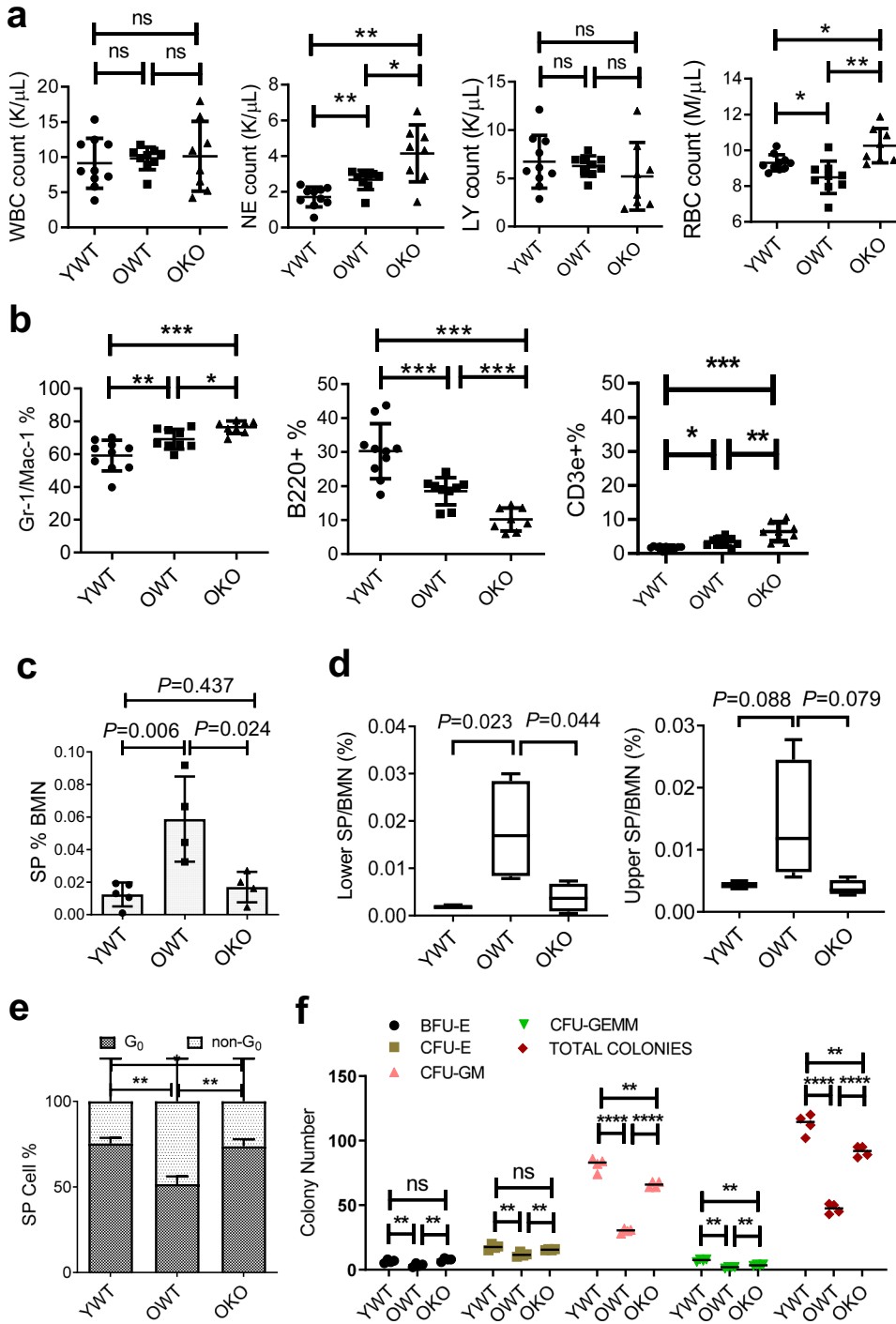

**Fig. 1 SIRT1 knockout inhibited HSC expansion and functional decline in aging BALB/c mice. a** Comparison of peripheral blood cell counts. WBC white blood cell, NE neutrophil, LY lymphocyte, RBC red blood cell. **b** Flow cytometry analysis of bone marrow cell lineages for myeloid (Gr-1/Mac-1$^+$), B (B220$^+$), and T (CD3e$^+$) cells among total nucleated cells. **c** Comparison of Lin$^-$ SP cells among bone marrow nucleated cells (BMN). **d** Lower and upper Lin$^-$ SP cells. **e** Cell cycle analysis of SP cells. **f** In vitro colony formation of bone marrow cells. OWT and OKO mice were in 16–20 months, and YWT in 2–3 months. CFU colony forming unit, GEMM granulocyte, erythrocyte, monocyte and megakaryocyte, CFU-GM colony-forming unit-granulocyte and monocyte, BFU-E burst-forming unit-erythroid, CFU-E colony-forming unit-erythroid. **$P < 0.01$; ****$P < 0.0001$; ns, not significant. Error bars represented one standard deviation.

OWT cells was shorter than normal BALB/c mice due to lethal irradiation as expected, the lifespan of the OKO recipients was significantly longer and closer to that of normal BALB/c mice (Fig. 2e). Therefore, OKO transplants yielded better hematological outputs and improved mouse survival in the 1st round BMT recipients.

**Sirt1 knockout inhibited age-dependent development of mouse B/M MPAL.** In the 2nd and 3rd round BMT, 90% of mice receiving OWT bone marrow cells succumbed to acute leukemia with over 20% of blasts in blood and bone marrow and died in a short period of time (Fig. 3). Splenomegaly and moderate to severe anemia were seen in all leukemic mice, and splenic

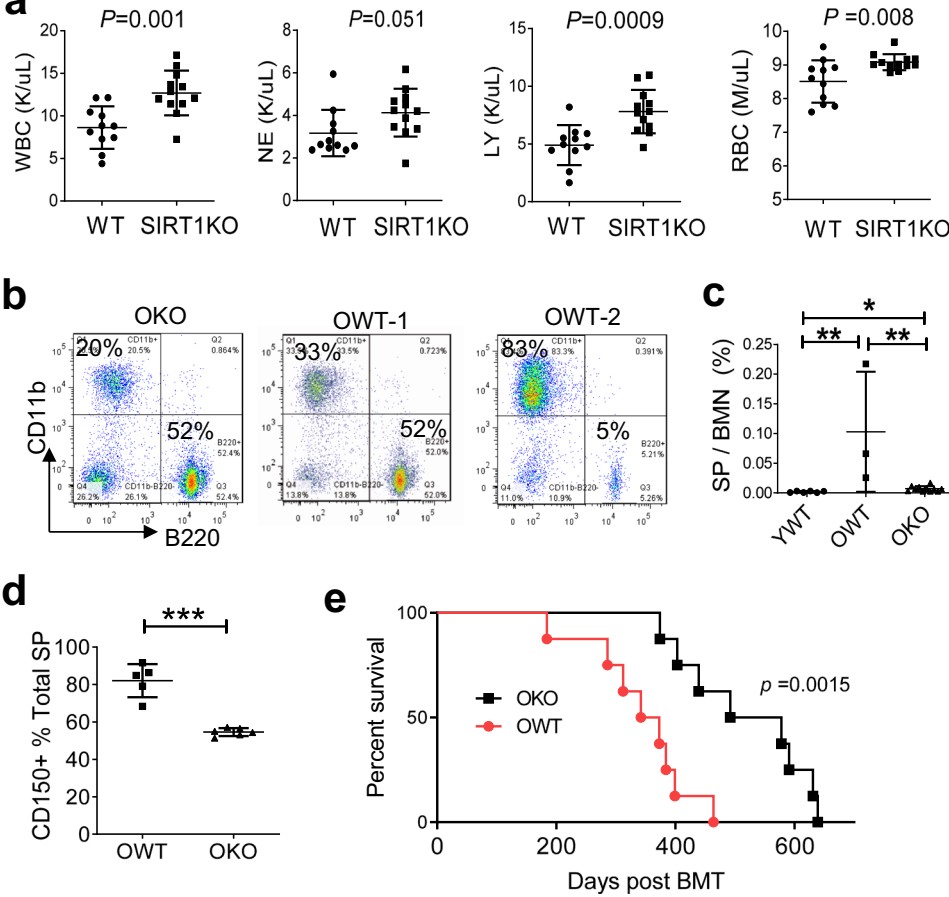

**Fig. 2 SIRT1 knockout inhibited the skewed differentiation of old HSCs in the 1st BMT recipients and improved mouse survival. a** Differential blood cell counts in 1st BMT recipients 8 months after transplantation with 14-month donor marrow (14-8 series). **b–d** Analysis of another BMT series 10 months after transplantation with 20-month-old donor marrow (20-10 series) to 8-week-old YWT recipients. **b** peripheral blood B220+ vs. CD11b+ cells; **c** bone marrow SP cell analysis; **d** CD150 fraction of bone marrow SP cells. CD150 is a marker for long-term HSCs. CD150+SP cells are known to increase with aging, but they were markedly reduced by Sirt1 knockout in the 1st round BMT recipients. **e** Kaplan–Meier survival analysis of 1st round OKO and OWT BMT recipients in another 20-X series in 5-month-old recipients. *P < 0.05; **P < 0.01; ***P < 0.001. Error bars represented one standard deviation.

lymphoma in ~20% of the mice. This phenotype occurred when the cumulative bone marrow age reached over ~36 months in the 2nd round BMT and ~31 months in the 3rd round BMT (Fig. 3c–e). The leukemia was transplantable with 100% disease penetrance in the secondary recipients. Strikingly, mice receiving OKO cells were much healthier and lived much longer (Fig. 3c–e). The incidents for acute leukemia were found in 18 out of 20 mice receiving OWT bone marrow cells in three independently completed cohorts in which all these mice died of diseases (excluding those found dead and decomposed), which contrasted with only 2 cases of acute leukemia in 29 mice receiving OKO cells in the cohorts (P < 0.0001). This phenotype has been observed with different original OWT donors and different ages of the 1st round BMT recipients (Fig. 3c–e). PCR analysis of *Sirt1* gene confirmed the effective engraftment of OKO cells in the recipients (Fig. 3f). Notably, when 2-month-old YWT mice were used as original donors with a short course of serial BMT, we did not observe such results[22,35], suggesting that the accumulated genetic and epigenetic changes in OWT HSCs during the course of bone marrow aging may contribute to the malignant phenotype, and Sirt1 loss inhibits malignant transformation.

The acute leukemia in OWT recipients morphologically appeared as AML (Fig. 3a). However, immunophenotyping revealed that the acute leukemia was B/M MPAL (in this case, it is also called biphenotypic acute leukemia): in the 14-19-4 BMT

series, leukemic blood cells expressed unusual intermediate levels of both lymphoid B220 and myeloid CD11b markers and appeared as CD43+CD19+CD127+ (Fig. 4a), suggesting a blockage of early B-cell development at the pro-B to pre-B stage. In the 14-8-14 BMT series, leukemia cells were B220 and CD11b double positive, and these B220+CD11b+ cells were CD43+CD19− (Fig. 4b), suggesting a blockage at the pre-ProB stage. Similarly, mice receiving wild type cells in the 3rd round BMT series also developed B/M MPAL characterized by B220+CD11b+ cells with a blockage at the pre-ProB stage. Collectively, these findings suggest an early B-cell differentiation arrest in B/M MPAL cells.

We next examined the changes of key B-cell differentiation genes in B/M MPAL. Differentiation of B-cells is initiated and maintained by *TCF3* gene encoding E2A protein that activates PAX5 and EBF1 to stimulate B-cell-specific gene expression[37]. Loss of *TCF3* by its fusion with *ZNF384* occurs frequently in B/M MPAL[15] and B-ALL[38,39]. Loss of Ebf1 and Pax5 causes growth arrest at early B cell progenitor stages in mice[40–42]. To examine the expression of these genes, we sorted B220+CD11b+ B/M MPAL cells and B220+CD11b− differentiated cells from OWT recipient mice and the corresponding cell fractions from OKO recipients. The residual B220+CD11b+ cells (<1%) in OKO mice were possibly from background staining and/or trace of such cells remained after MPAL inhibition. For a comparison,

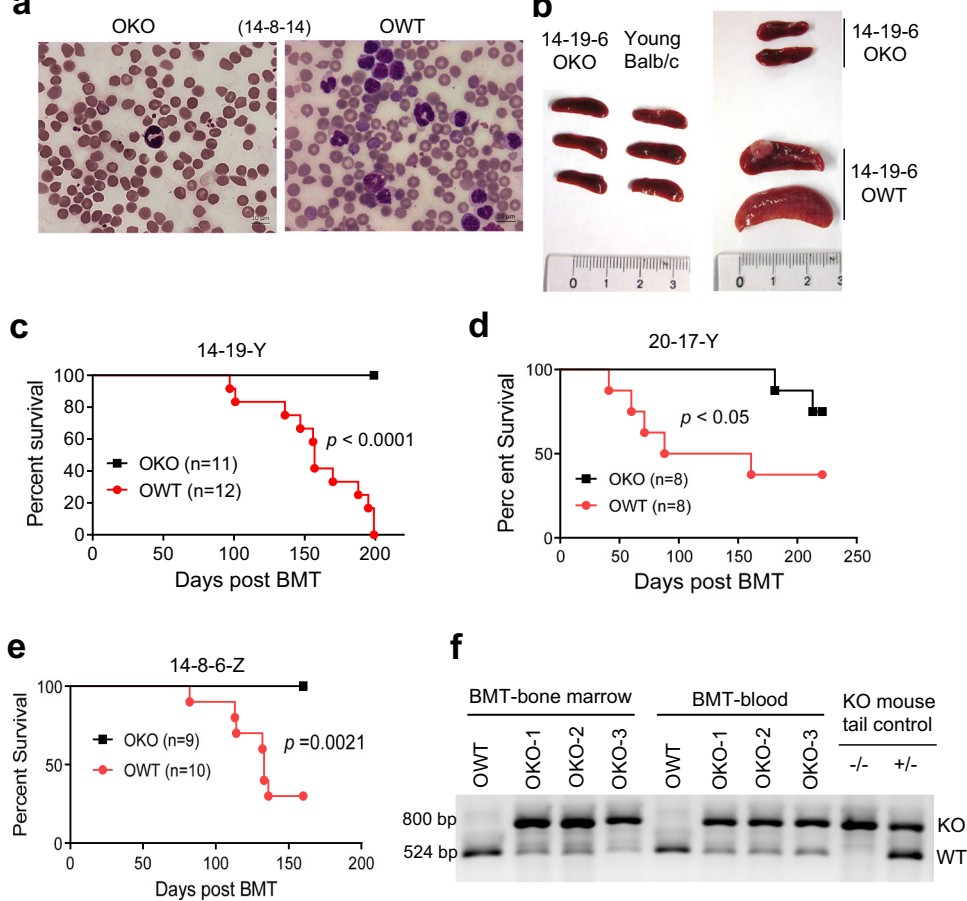

**Fig. 3 SIRT1 knockout inhibited spontaneous acute leukemia in 2nd and 3rd BMT recipients. a** Representative blood smear of 2nd BMT mice (14-8-14 series). **b** Splenomegaly in 14-19-6 BMT recipients with WT, but not SIRT1 knockout, donor cells. Noted a lymphoma nodule on one WT spleen. **c–e** Survival curves of 2nd BMT recipients (14-19-Y series in **c** and 20-17-Y series in **d**) and 3rd BMT recipients (14-8-6-Z in **e**). **f** PCR confirmation of donor cell identity in bone marrow and blood of 2nd BMT recipients. Residual WT DNA seen in OKO recipients was due to minor contamination of recipient tissues.

B220⁻ cells (CD11b⁺ and CD11b⁻) were also sorted for analysis. We found that *Tcf3*, *Pax5* and *Ebf1* mRNAs were significantly lower in B/M MPAL cells than in differentiated B220⁺CD11b⁻ cells in OWT mice (Fig. 4c). Sirt1 loss significantly increased their expression in both B220⁺CD11b⁺ and B220⁺CD11b⁻ cells, but not in B220⁻ cells (Fig. 4c), indicating B-cell-specific regulation of these genes by Sirt1. Ebf1 and Pax5 are essential for B-cell V(D)J recombination. We examined V(D)J recombination in B220⁺CD11b⁺ B/M MPAL cells vs. other cell fractions by PCR assays as described[42]. The germline D-J allele was near fully retained in B220⁺CD11b⁺ MPAL cells and partially retained in B220⁺CD11b⁻ cells in OWT recipients; however, this was not observed in OKO recipients (Fig. 4d). Together, these results revealed a severe defect of V(D)J recombination in B/M MPAL and a partial defect in OWT B cells, which was inhibited by Sirt1 knockout that increased the expression of B-cell differentiation regulators.

**SIRT1 regulation of B-cell differentiation genes and its effects on human B/M MPAL**. We next determined potential roles of SIRT1 in human B/M MPAL. SIRT1 is rarely mutated in human cancer[43], but is subjected to multiple layers of regulation, including transcription, mRNA stability, posttranslational modifications, and enzymatic activity[29]. In CML stem/progenitor cells SIRT1 mRNA and protein are both upregulated[22,23]. However, in FLT3-ITD AML cells SIRT1 protein, but not mRNA, is upregulated[24]. Given that MPAL is treated more favorably with ALL therapies, we

included B-ALL cells in the study. We found that SIRT1 protein was over-expressed in primary human adult B/M MPAL and B-ALL blasts, as compared to normal adult B cells (Fig. 5a). Similarly, SIRT1 protein levels were increased in a B/M MPAL cell line (MV4-11) and B-ALL cell lines (Sup-B15, RS4-11 and REH), as compared to normal human CD19⁺ B cells (Fig. 5b). MV4-11 is a B/M MPAL cell line that was established from a biphenotypic B/myelomonocytic leukemia patient[44,45], even though it is often used as an AML cell line. We did not detect an increase of SIRT1 mRNA in B/M MPAL and B-ALL cells as compared to normal B cells, and Sirt1 mRNA levels in B/M MPAL cells lied between that of normal B and myeloid cells, for which normal B cells had much higher Sirt1 mRNA than myeloid cells (Supplementary Fig. 3).

To examine if SIRT1 may regulate human B-cell differentiation genes, we knocked down SIRT1 in human B/M MPAL and B-ALL cell lines. We found that *TCF3, PAX5* and *EBF1* mRNA expression was all increased after SIRT1 knockdown in these cell lines (Fig. 5c). Western blot confirmed that E2A protein encoded by *TCF3* gene was increased after SIRT1 knockdown (Fig. 5d). These results indicate a conserved role of SIRT1 in regulating mRNA expression of early B-cell differentiation genes in both mice and humans, whose dysregulation may contribute to B/M MPAL development.

E2A protein is acetylated and activated by CREBBP/EP300, which enhances E2A nuclear localization and transcriptional activity[46]. By co-expressing the large E2A splice variant E47 with Flag-SIRT1 in 293T cells, we found that E47 and SIRT1 were

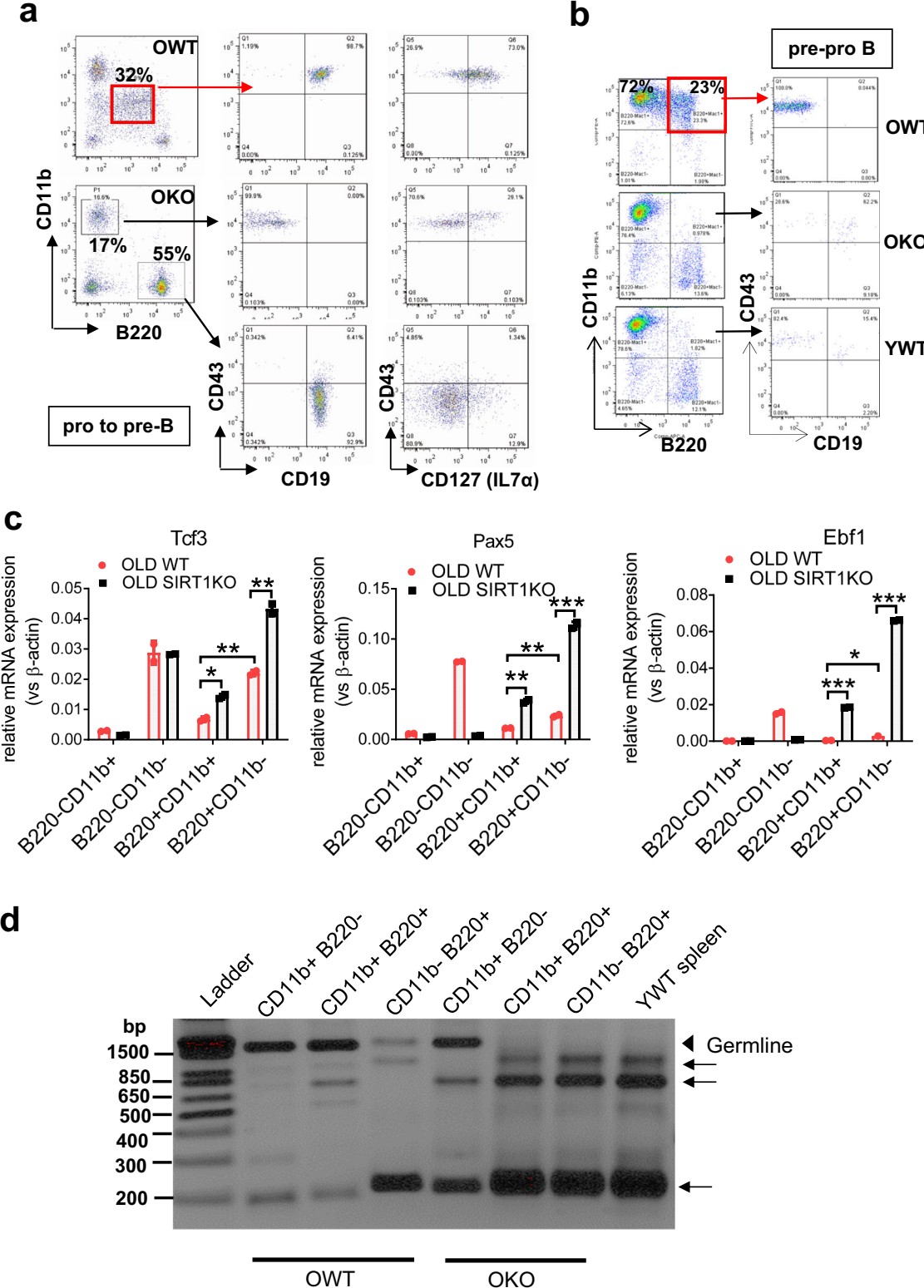

co-immunoprecipitated mutually, indicating interaction of the two proteins (Supplementary Fig. 4a, b). Further, E2A acetylation by CREBBP/EP300 was effectively reversed by SIRT1 (Supplementary Fig. 4c). These data suggest that SIRT1 can interact with and modify E2A posttranslationally. However, the precise role of SIRT1 deacetylation of E2A in B cell differentiation remains to be delineated. Functionally, SIRT1 knockdown inhibited growth and induced the apoptosis of MV4-11 and Sup-B15 cell lines, and

sensitized them to chemotherapeutic agent camptothecin and oxidative stress inducer hydrogen peroxide (Fig. 5e, f), in line with previous reports[24,25].

To further corroborate the roles of SIRT1 in human leukemia, we carried out CRISPR (clustered regularly interspaced short palindromic repeats) gene editing to knockout SIRT1 in human primary B/M MPAL and B-ALL cells. The primary cells were expanded and maintained in growth factor-supplemented media.

**Fig. 4 Aging BALB/c mice with serial BMT developed MPAL that was inhibited by Sirt1 knockout. a** Immunophenotyping of blood immature cells in 14-19-4 BMT mice. Note the unusual immature cell population with intermediate B220 and CD11b expression. **b** Immunophenotyping of acute leukemia in 14-8-14 BMT mice. Note high percentage of B220$^+$CD11b$^+$ cells and absence of mature B cells in bone marrow of mice receiving WT cells. **c** Realtime RT-PCR analysis of mouse B-cell differentiation genes in B/M leukemia cells and non-leukemia fractions from OWT and OKO BMT recipient mice. **d** PCR analysis of V(D)J recombination in 14-8-14 series B/M MPAL cells. Myeloid cells (CD11b$^+$B220$^-$) in both OWT and OKO retained most of the unrecombined germline allele indicated by the arrowhead. YWT spleen displayed fully recombined D-J alleles indicated by the arrows as a result of normal B-cell development, and OKO B220$^+$ cells (either CD11b$^-$ or Cd11b$^+$) exhibited the same full D-J recombination patterns as YWT spleen. In contrast, OWT B220$^+$CD11b$^+$ leukemic cells retained most of the D-J germline allele and OWT B cells (B220$^+$CD11b$^-$) did not undergo full D-J recombination like YWT spleen and retain a residual amount of the germline allele. *$P < 0.05$; **$P < 0.01$; ***$P < 0.001$. Error bars represented one standard deviation.

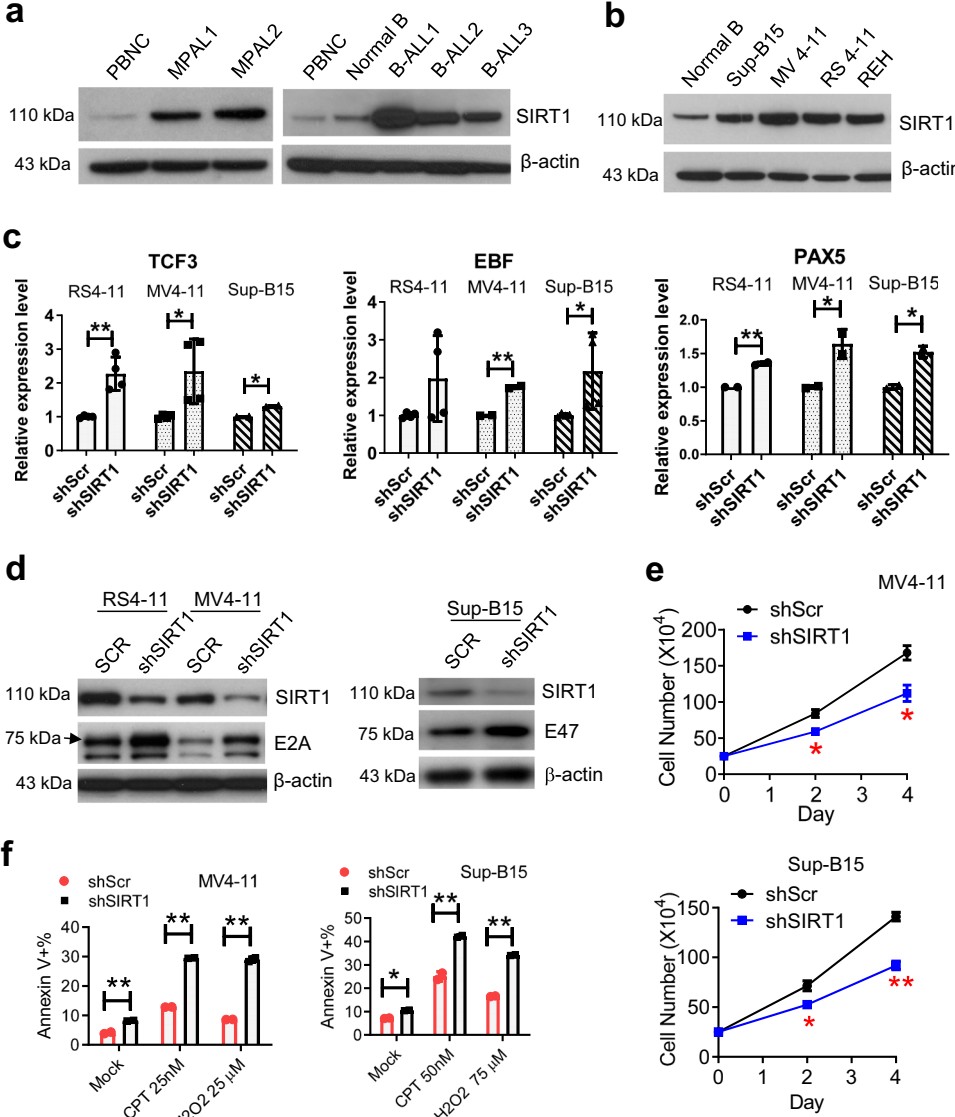

**Fig. 5 SIRT1 gene expression in human MPAL cells and regulation of human B-cell-specific genes. a** SIRT1 protein expression in human adult B/M MPAL and B-ALL. Normal B: purified normal human B cells that contained >95% CD19+ cells. PBNC: total peripheral blood nucleated cells. Ficoll-enriched primary leukemia samples contained 85–95% blasts. The age of patients was ranged from 27 to 60. **b** SIRT1 protein expression in MPAL and B-ALL cell lines. **c** Realtime RT-PCR analysis of TCF3, EBF and PAX5 expression in MPAL and B-ALL cell lines after SIRT1 knockdown. shScr scrambled shRNA. **d** Western blot analysis of E2A or the large splice variant of E2A, i.e. E47, upon SIRT1 knockdown. **e** Effects of SIRT1 knockdown on growth of MV4-11 and Sup-B15 cells. **f** Effects of SIRT1 knockdown on cell apoptosis in the absence and presence of camptothecin (CPT) or $H_2O_2$ treatment. *$P < 0.05$; **$P < 0.01$. Error bars represented one standard deviation.

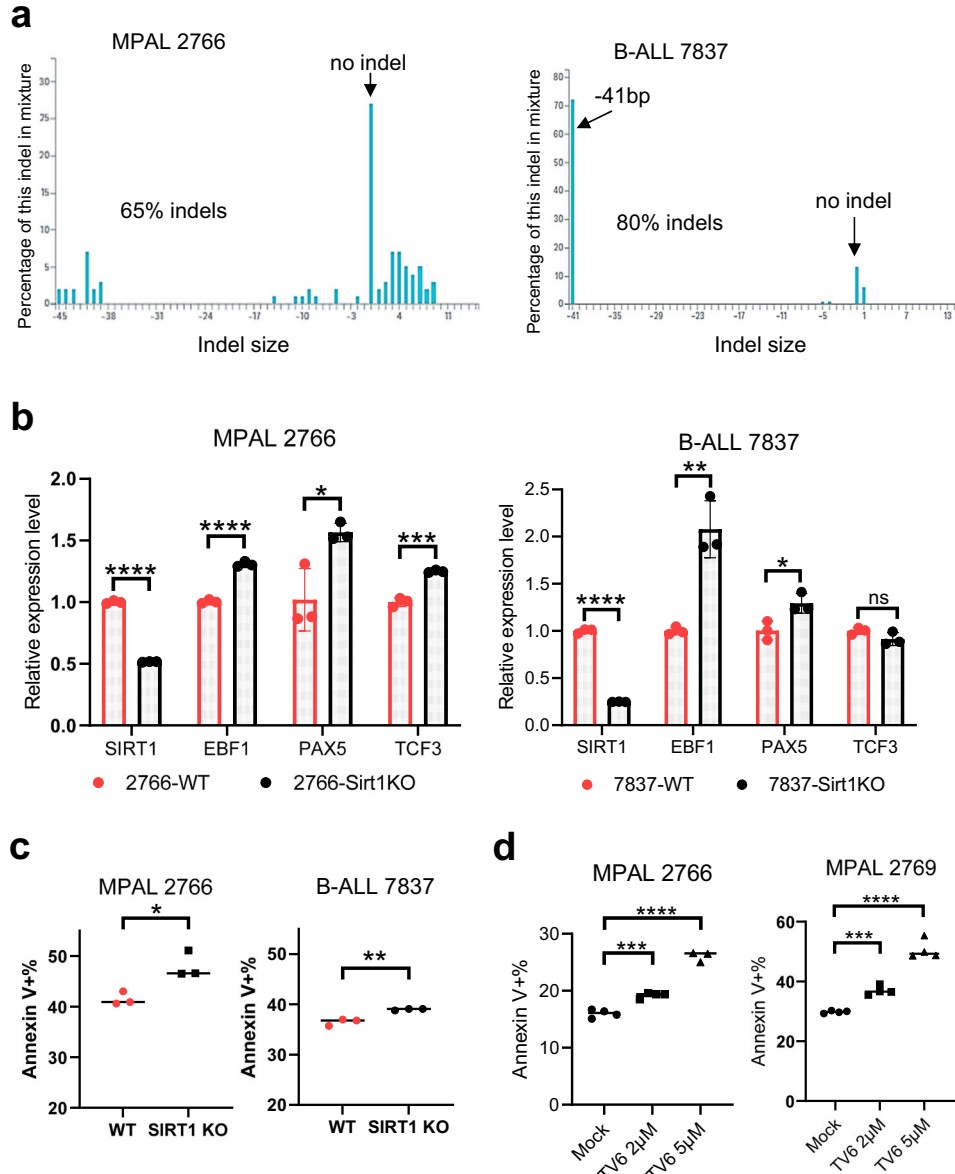

**Fig. 6 Effects of SIRT1 knockout and inhibition on human primary leukemia cells. a** CRISPR-mediated gene knockout of SIRT1 in human primary B/M MPAL and B-ALL cells. Indel size distribution and percentage of each indel at day 3 after electroporation were shown. There were 18% inframe indels in MPAL cells whereas inframe indels in B-ALL cells were below detection limit. **b** RT-qPCR analysis of SIRT1 and its target genes at day 3 after electroporation in MPAL and B-ALL cells. Wild type (WT) were mock knockout samples undergone the same procedures without sgRNAs. SIRT1 RT-qPCR primers were located at the exon 4–5 junction. **c** Apoptosis analysis of primary leukemia cells after SIRT1 KO at day 6. **d** Apoptosis analysis of two primary adult B/M MPAL samples 3 days after treatment with SIRT1 inhibitor tenovin-6 (TV-6). *$P < 0.05$; **$P < 0.01$; ***$P < 0.001$; ****$P < 0.0001$. Error bars represented one standard deviation.

Two sgRNAs targeting SIRT1 exon 2 were electroporated into the cells using a ribonucleoprotein (RNP) system. Three days after electroporation, *SIRT1* gene KO was detected with 65% indels in MPAL and 80% indels in B-ALL (Fig. 6a), which was maintained at Day 6. Indels in B-ALL cells were predominated by a 41 bp deletion whereas indels in MPAL cells varied in size (Fig. 6a). Notably, 18% of indels in MPAL were inframe whereas no inframe indels were detected in B-ALL cells at Day 3. In line with DNA KO, *SIRT1* mRNA was reduced more in B-ALL than in MPAL cells (Fig. 6b). *TCF3, PAX5* and *EBF1* mRNAs were increased after SIRT1 KO in MPAL cells, whereas *PAX5* and *EBF1* mRNAs were increased in B-ALL cells (Fig. 6b). A moderate but significant increase of apoptosis was detected after SIRT1 KO in MPAL and B-ALL cells (Fig. 6c), with the former

more obvious, suggesting that MPAL is likely more sensitive to SIRT1 KO. The moderate increase of apoptosis was partly due to high background cell death caused by electroporation, which obscured the flow cytometry analysis of annexin V. We then treated two human primary MPAL samples with SIRT1 inhibitor tenovin-6 [22,23]. This treatment led to a significant increase of MPAL apoptosis in a dose-dependent manner (Fig. 6d). Altogether, our data suggest that SIRT1 is over-expressed in human B/M MPAL and B-ALL cells, suppresses the expression of B-cell differentiation genes, and regulates their growth and survival.

**Sirt1 knockout inhibited protein synthesis and metabolic activation in aging HSCs.** The founding lesions of human MPAL arise in primitive hematopoietic progenitors, which may alter

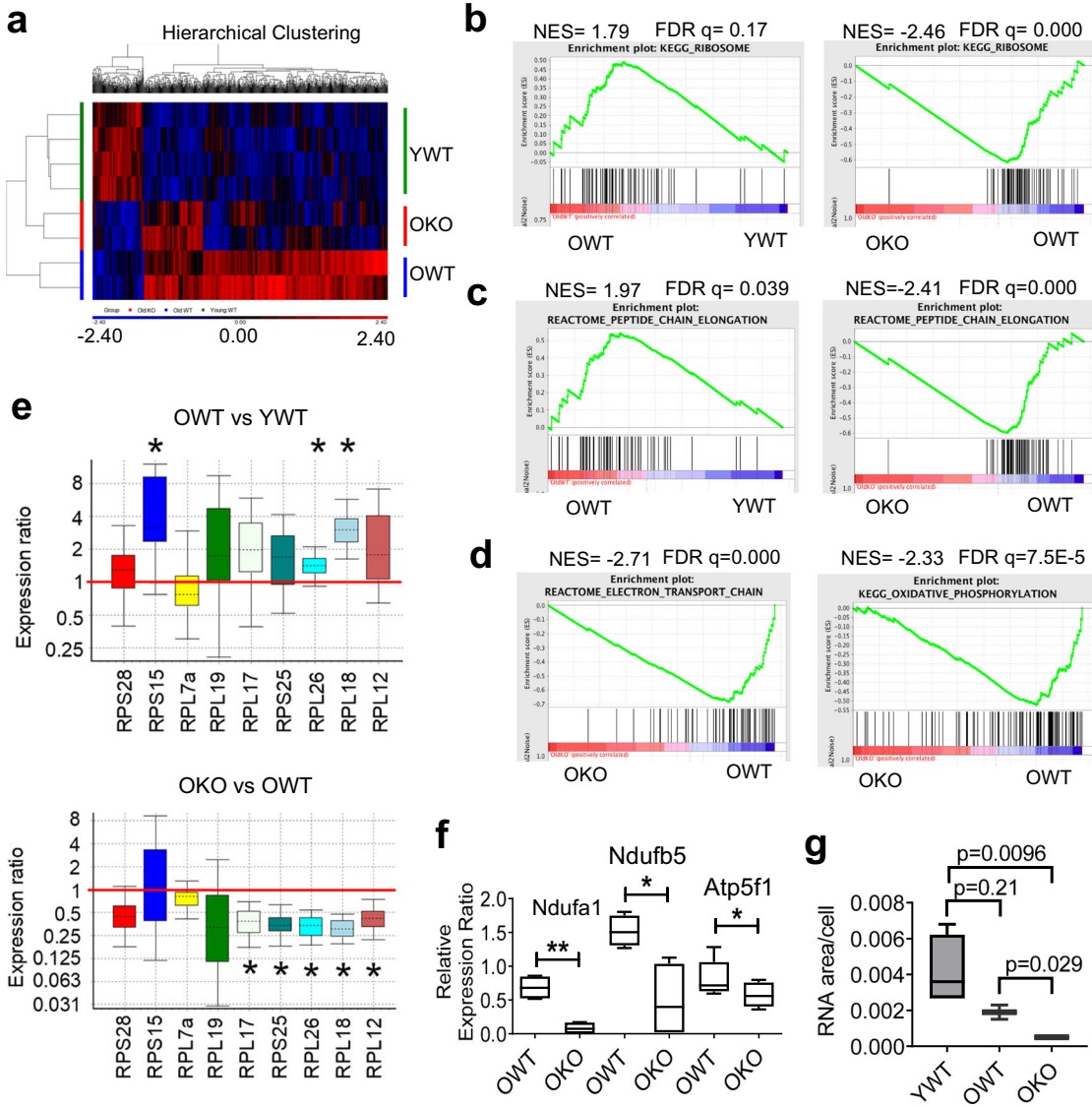

**Fig. 7 Sirt1 knockout reduced age-dependent changes of molecular pathways in aging HSCs. a** Hierarchical clustering of microarray gene expression in SP cells. **b–d** GSEA plots for ribosomal genes (**b**), peptide chain elongation (**c**), and electron transport chain/OXPHOS (**d**). The method for ranking genes was the default "Signal2Noise," and the scoring scheme was also the default "weighted". GSEA plots were ranked by NES and the full sets of results were provided in Supplementary Data 1 and 2. **e** OWT/YWT and OKO/OWT gene expression ratio of ribosomal genes analyzed by real time RT-PCR. **f** Real time RT-PCR analysis of top ranked OXPHOS genes in OWT and OKO HSCs. **g** Analysis of 18S + 28S rRNA of HSCs by Agilent Bioanalyzer. *$P < 0.05$; **$P < 0.01$. Error bars represented one standard deviation.

their differentiation regulation[15]. In our data above, mouse B/M MPAL phenotype was dependent on serial BMT and cumulative age of bone marrow cells, most importantly HSCs. These lent support to that HSC aging and possibly the alterations acquired in aging HSCs, both genetic and non-genetic, may set the stage for adult B/M MPAL development. To better understand the changes in aging HSCs and the impact of Sirt1 knockout, we carried out gene expression profiling of aging HSCs by microarray analysis with Affymetrix Mouse Genome 430 2.0 gene chips for Lin⁻SP HSCs. We found that many genes were activated in OWT HSCs as compared to YWT, which were reversed in OKO (Fig. 7a). Although this seems counterintuitive to Sirt1 functions as a histone deacetylase and transcriptional repressor, we found that OWT hematopoietic stem/progenitor cells had increased histone H4 acetylation as compared to YWT cells, which was partially reversed in OKO stem/progenitor cells (Supplementary Fig. 5).

Multiple pathways involved in protein synthesis control including ribosomes (both 40S and 60S), peptide chain elongation, and viral mRNA translation were the most prominent pathways enriched in OWT over YWT HSCs, along with several other signaling pathways including TGFβ and MAPK pathways (Fig. 7b, c and Supplementary Fig. 6, and a full list of pathways and rankings was provided in Supplementary Data 1). With Sirt1 knockout, protein synthesis/ribosomes and electron transport chain (ETC)/oxidative phosphorylation (OXPHOS) were among the top pathways that were inhibited (Fig. 7b–d and Supplementary Fig. 7, and a full list of pathways and rankings was provided in Supplementary Data 2). Accompanied with these changes, G1-S transition, WNT signaling, degradation of p27/p21, insulin synthesis/secretion, and proteasome were also down-regulated by the knockout (Supplementary Fig. 7). In addition, upregulated pyruvate metabolism, TCA cycle, and fatty acid metabolism in OWT HSCs were downregulated in OKO HSCs (Supplementary

Data 1 and 2), further indicating a reversal of metabolic activation in OKO HSCs. We confirmed the mRNA expression changes of top ranked ribosomal protein genes (Fig. 7e) and OXPHOS genes such as complex I factors Ndufa1 and Ndufb5 (Fig. 7f). The rRNA synthesis is another key aspect of protein synthesis control that is regulated differently from ribosomal protein mRNAs[47]. We found that rRNA level was moderately down in old HSCs; but Sirt1 knockout significantly reduced it (Fig. 7g), in line with the inhibition of overall protein synthesis pathways. Protein synthesis is one of the most energy-demanding cellular processes, and is highly regulated in HSCs[48]. Inhibition of protein synthesis suppresses the proliferation of leukemic progenitor cells[48], and may curb the demand for energy of aging HSCs as they shift from glycolytic to oxidative metabolism[49,50]. Inhibition of protein synthesis pathways by Sirt1 knockout in old HSCs is thus in line with the reversal of metabolic activation in these cells.

A group of genes displayed Sirt1-independent changes, especially those towards the left side on the gene heatmap (Fig. 7a). To identify these genes, we used Venn diagram to overlap genes that were upregulated or downregulated in the analyses of both OWT vs YWT and OKO vs YWT (Supplementary Fig. 8). We identified 111 upregulated and 71 downregulated probesets, and among them 156 were annotated genes (Supplementary Data 3). The pathways for the upregulated genes in old mice included TNFα signaling via NF-kB, estrogen responses, hypoxia, etc; and the pathways for the downregulated genes in old mice included E2F targets, G2-M checkpoint, etc. (Supplementary Data 4).

**Sirt1 regulated multiple key transcriptional factors in aging HSCs.** To identify lead candidate genes that may be regulated by SIRT1, we performed 4-way Venn diagram analysis of the microarray data sets. This identified 250 genes that were up in OWT vs. YWT but down in OKO vs. OWT HSCs, and 10 genes that were down in OWT but up in OKO HSCs (Fig. 8a). Among the top ranking genes, we confirmed that *Trim26* (ranked #2 from the 250 gene list, Supplementary Data 5) was up in OWT HSCs but down in OKO (Fig. 8b), This analysis also identified several important transcriptional factors in oncogenesis and HSC maintenance that were down-regulated by Sirt1 knockout including HoxA5, Gsk3b, c-Myb and Dnmt1 (Fig. 8b and Supplementary Data 5). *HoxA5* along with *HoxA9* is highly expressed in human ALL with MLL gene translocation and associated with poor prognosis of the disease[51,52]; *Gsk3b* and *c-Myb* enhance oncogenic potential of aging HSCs[53,54], and *Dnmt1* mediates silencing of genes marked by bivalent chromatin domains in leukemic progenitor cells[55]. C-Myc was upregulated in OWT HSCs (Fig. 8b). Although c-Myc mRNA was not changed by Sirt1 KO, Sirt1 deficiency can inactivate c-Myc protein functions in cancer[56]. Finally, among the genes that were activated by SIRT1 knockout, we confirmed that *Tcf3* was up in OKO HSCs (Fig. 8b). Expression of Tcf3 in HSCs promotes maintenance of long-term HSCs and development of lymphoid-primed multipotent progenitors[57–59].

The roles of the top-ranking gene *Trim26* in HSCs and leukemia are unknown. *Trim26* is expressed in multiple tissues in mammals and is highly conserved between mice and humans[60]. Trim26 belongs to a family of tripartite motif (TRIM)-containing proteins that have E3 ubiquitin ligase activities and are involved in innate immunity and regulation of viral infection[61]. We validated the changes of Trim26 expression in aging HSCs in independent cohorts of aging Sirt1$^{+/+}$ and Sirt1$^{-/-}$ HSCs (Fig. 8c). Trim26 expression was also increased in the spleen from OWT and reduced in OKO mice, but such changes did not occur in the heart (Fig. 8d). Trim26 participates in epigenetic reprogramming of mouse embryonic fibroblasts (MEFs) to

induced pluripotent stem cells (iPSCs) by mediating ubiquitin-dependent degradation of plant homeodomain finger protein 20 (PHF20)[62]. We found that Sirt1 knockout reduced Trim26 expression in MEFs, but increased PHF20 (Fig. 8e). We performed Venn diagram analysis of our microarray data for genes affected by Sirt1 knockout in HSCs and the published data of PHF20 target genes in mouse iPSCs[62]. We found that 41% of the downregulated genes by Sirt1 knockout overlapped with PHF20 target genes, and 36% of 222 annotated genes from the 250-gene list that were up in OWT but down in OKO overlapped with PHF20 targets (Fig. 8f, g). Therefore, Trim26–PHF20 pathway may in part contribute to the effect of Sirt1 KO in old HSCs, which warrants further investigation. Together, these data suggest that Sirt1 knockout affects multiple transcriptional factors crucial for leukemogenesis and HSC functions, and among them Trim26 is a new downstream effector of Sirt1 in aging HSCs.

## Discussion

In this study, we used old mouse bone marrow cells for serial BMT to investigate HSC aging and the impact of Sirt1 knockout. This study led to several interesting discoveries. First, Sirt1 knockout unexpectedly deterred HSC aging in the knockout mice and the 1st round BMT recipients before MPAL development. Sirt1 knockout improved the maintenance of HSC quiescence, reduced myeloid skewing, and prolonged lifespan of the 1st round BMT recipients. The improvement of aging HSC functions by Sirt1 knockout was corelated with the suppression of genome-wide activation of numerous genes in aging HSCs, reversing their expression patterns closer to that of young HSCs. The genes and pathways regulating protein synthesis and oxidative metabolism were the prime targets of Sirt1 knockout. These results are consistent with previous reports that inhibition of OXPHOS overcomes metabolic shift from glycolytic in young HSCs to oxidative in old HSCs[49,50], and that low protein synthesis is necessary for HSCs to maintain self-renewal and inhibits leukemogenesis[48]. The suppression of metabolic activation and protein synthesis pathways by Sirt1 knockout may thus improve aging HSC functions and inhibit their transformation into leukemia stem cells for MPAL development down the road. Our finding is opposite to the conventional wisdom that loss of Sirt1 would activate gene expression since it is a known histone deacetylase and transcriptional repressor. This is likely caused by differential roles of Sirt1 in young and old HSCs. It is shown that SIRT1 is redistributed on chromatin to facilitate DNA repair and change global gene expression during aging[63] and that in response to stress and DNA damage, SIRT1 is recruited to maintain histone H4 acetylation and open chromatin for repair[64]. Therefore, Sirt1 may behave oppositely in old HSCs vs. young HSCs in response to increased DNA damage and stress in old HSCs[65], which results in reduced histone acetylation and repression of gene expression upon the knockout in old HSCs.

We identified several downstream transcriptional factors including Dnmt1, HoxA1, Gsk3b and c-Myb that were targeted by Sirt1 in aging HSCs and are known for regulating HSC functions and oncogenesis. We found Trim26 as a new downstream effector of Sirt1 more strictly in hematological organs. Trim26 is an E3 ubiquitin ligase involved in epigenetic regulation of stem cell fate through Trim26-PHF20 pathway[62]. Trim26-PHF20-regulated genes partially overlapped with Sirt1-regulated genes, suggesting a potential contribution of this pathway. Trim26 has been implicated in innate immunity and viral infection[66], but its roles in HSCs and leukemia are unknown and further investigations are warranted.

Our result of improving old HSC functions by Sirt1 knockout is in stark contrast to Singh et al. showing that Sirt1 conditional

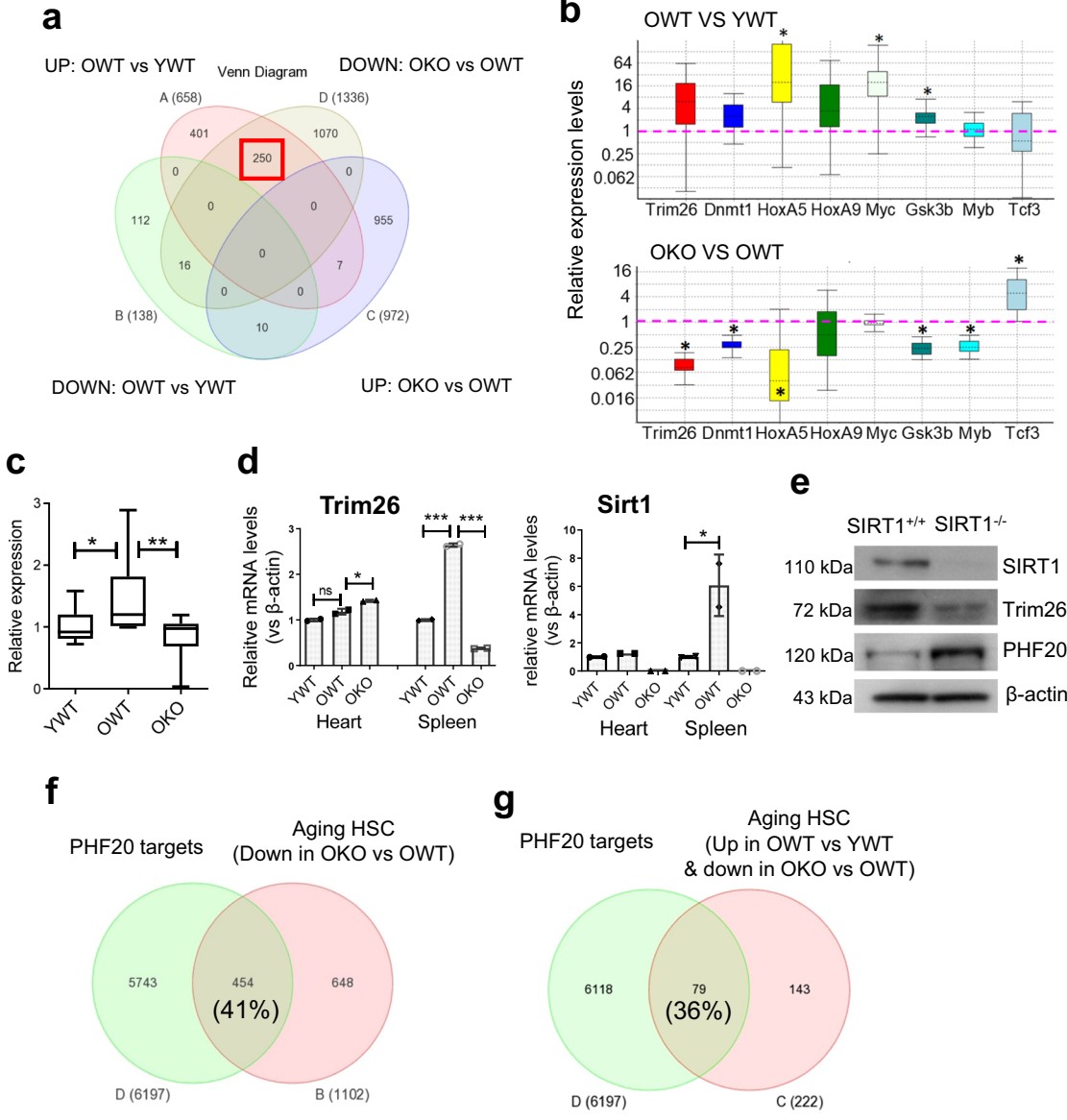

**Fig. 8 SIRT1 knockout impacted multiple downstream effectors in aging HSCs. a** Venn diagram analysis identified 250 genes that were up in OWT compared to YWT, but were down-regulated in OKO HSC cells. **b** Realtime RT-PCR validation of key genes identified in **a**. Geometric averaging of multiple control genes (β-actin, 18S RNA and CD45) was used to normalize data. **c** Trim26 mRNA in purified HSCs in independent aging cohorts ($n = 7$ each). **d** Trim26 and Sirt1 mRNA in spleen and heart using unfractionated cells. **e** Western blot analysis of MEF cell lysates. **f**, **g** Overlap of PHF20 target genes with genes down in OKO over OWT HSCs (**f**) and 222 annotated genes from 250 genes that were up in OWT vs YWT but down in OKO (**g**). *$P < 0.05$; **$P < 0.01$; ***$P < 0.001$. n.s. not significant. Error bars represented one standard deviation.

knockout compromised HSC functions in younger mice[67]. However, the findings of Singh et al. were confounded by the toxicity to mouse bone marrow caused by tamoxifen and early mouse death after vav-iCre expression as discussed previously[68]. Using a similar approach, Rimmele et al. reported aging-like phenotypes of hematopoietic stem/progenitor cells in Sirt1 conditional knockout without inducing bone marrow toxicity[69]. Their phenotypes are mostly different from ours and the influence of tamoxifen-induced Cre expression could not be excluded. Besides, their study was conducted in young mice that had not undergone physiological aging. Notably, increased HSC cell cycling in their study was also found in our young Sirt1$^{-/-}$ mice[35]; however, it did not translate into the real aging phenotype here. Furthermore, HSC functions were not compromised in a recent study with Mx1-Cre-mediated conditional Sirt1 knockout mice[30]. The inconsistent results of Sirt1 conditional knockout

mice could be due to different manipulation of Cre expression that might differently affect mouse bone marrow microenvironment, which has also confounded some studies of HSC-niche interaction[70]. Alternatively, the discrepancy may reflect distinct roles of Sirt1 in young vs. old HSCs as we discuss above. Another possible contributing factor would be the mouse strain background. The early developmental lethality after Sirt1 knockout differs by mouse strains[71]. Although the effect of strain background on HSC aging remains to be formally determined, we found that SIRT1 inhibitors stimulated methylcellulose colony formation of aging mouse HSCs from both BALB/c and C57BL/6 mice, suggesting that the effect of Sirt1 knockout on HSC aging is likely not strain-dependent. With that said, however, it is possible that the hypomorphic *Cdkn2a* (*p16$^{Ink4a}$*) allele in BALB/c mice[72,73] may play a role in the phenotypes we observed. Cdkn2a-mediated cell senescence is implicated in aging of solid

organs and their stem cells, but not HSCs[74,75], suggesting the unique regulation of HSC aging. On the other hand, *CDKN2A* loss either by genetic alterations or epigenetic gene silencing is one of the most frequent events in cancer, which allows pre-cancerous cells to bypass senescence for tumorigenesis[74]. Consistently, genetic mutations and promoter hypermethylation of *CDKN2A* are frequent in human adult MPAL[14]. Hypomorphic *Cdkn2a* of BALB/c mice may facilitate MPAL development in our model discussed further below. Based on our results, we propose that SIRT1 promotes HSC aging in the presence of defective Cdkn2a and facilitates age-dependent oncogenic transformation of HSCs for MPAL development. This may account for the difference of this study from others that SIRT1 promotes healthy aging. It would be interesting to further determine how these two genes may interplay for HSC aging and MPAL development in the future.

We found that B/M MPAL occurred in 90% BMT recipients when the cumulative age of donor cells reached over 31 months. To our knowledge, this is the first mouse model producing such a high frequency of MPAL, which morphologically and immuno-phenotypically mimics the human counterpart and carries no artificial transgenes. Human MPAL is a difficult disease to study and currently there is no other mouse model available since introducing MPAL-related genetic lesions into mice typically results in AML, ALL or myelodysplastic syndrome, but not MPAL[76–78]. Although additional molecular characterization is needed to further validate our model for the human disease, this model provides the first step for our understanding of aging impact on dysregulation of HSCs and their priming for MPAL development in vivo and lays a foundation for further innovation and improvement of the disease modeling. Our results support that MPAL is a stem/progenitor cell disease and HSC aging plays an important role in the disease progression, which is consistent with a recent report of human MPAL in that the founding lesions and the cell of origin prime MPAL cells for lineage promiscuity[15].

Our discovery that Sirt1 knockout inhibited mouse MPAL development is exciting given that there is no targeted therapy currently available for MPAL. Although SIRT1 has been shown to be an oncogene[22–25,30,35] or a tumor suppressor[79] in other types of leukemia, its role in MPAL is unknown. We found that Sirt1 specifically regulated B-cell differentiation program. Loss of Sirt1 stimulated Tcf3, Pax5 and Ebf1 expression in aging mouse B cells and promoted V(D)J recombination, which in-turn suppressed B-cell differentiation arrest for mouse MPAL development. The role of SIRT1 in regulating B-cell differentiation genes was conserved in mouse and human cells. SIRT1 protein was over-expressed in human B/M MPAL cells, and SIRT1 gene knockout or inhibition by a small molecule inhibited their growth and survival. Our results place SIRT1 as a potential novel therapeutic target for B/M MPAL, which has important translational implications.

In conclusion, we demonstrate that deletion of Sirt1 inhibits mouse HSC aging and age-dependent development of B/M MPAL. SIRT1 is over-expressed in human MPAL and SIRT1 inhibition suppresses growth and survival of human MPAL. These results raise the possibility of targeting SIRT1 for inhibiting HSC aging and treatment of MPAL. Our study also establishes a new approach for modeling human B/M MPAL in immune-competent mice, which will facilitate future mechanistic studies of the disease and in vivo testing of novel therapies.

## Methods

**Cell lines and primary human cells**. Human leukemia cell lines were purchased from American Type Culture Collection, Manassas, VA 20110, USA, or German Collection of Cell Cultures, Braunschweig, Germany. Primary human leukemia cells were provided by Dr. James You from MD Anderson Cancer Center, and these samples were deidentified before distribution for the study. All experiments were conducted under protocols approved by Institutional Biosafety Committee of City of Hope.

**Mice**. All animal experiments were conducted under a protocol approved by the City of Hope Institutional Animal Care and Use Committee. For serial BMT experiments, BALB/c mice (Taconic) at the age of 7–8 weeks were used as recipients unless indicated. The SIRT1 knockout mice used in the study were back-crossed to BALB/c strain (Taconic) for at least nine generations. Sirt1[−/−] mice were produced by breeding Sirt1[+/−] mice using a consecutive fostering protocol[35]. All mice were housed and maintained in accordance with the Guide for the Care and Use of Laboratory Animals at the AAALAC International accredited specific-pathogen-free Animal Resources Center within the Center for Comparative Medicine at City of Hope, and fed ad libitum with LabDiet 5053 standard mouse chow and provided reverse osmosis (RO) treated water. The animal room temperature was held within a range of 68 to 75 °F, humidity was 30–70%, and light cycle was 12 h light:12 h dark. The 16–20-month-old mice were used as aging donor mice in the studies.

**Mouse hematopoietic lineage analysis, hematopoietic stem/progenitor cell isolation and analysis**. Mouse peripheral blood cell counts were measured by Hemavet (Drew Scientific). For flow cytometry lineage analysis, peripheral blood or bone marrow nucleated cells were stained with an antibody cocktail containing anti-CD11b and anti-Mac1 for myeloid cells, anti-B220 for B cells and anti-CD3e for T cells. For bone marrow stem cell analysis and sorting by side population, cells were stained with 5 μg/mL Hoechst 33342 at 37 °C in the DMEM+ media (DMEM with 2% heat-inactivated fetal bovine serum and 1 mM HEPES) in a density of 1 million cell per mL for 90 min exactly, followed by depleting lineage-positive cells using EasySep Stem Cell Enrichment Kit (StemCell Technologies). Then the enriched cells were labeled with PE-conjugated lineage antibody cocktail and other cell markers. Mouse hematopoietic stem cells in the BABL/c strain were analyzed by flow cytometry using the PE-Lin/Side Population/APC-CD150 combination, as described previously[35]. Flow cytometry analysis and cell sorting were carried out at the City of Hope Analytic Cytometry Core.

To analyze cell cycle of hematopoietic progenitor cells and stem cells, bone marrow cells were stained according to the above-mentioned procedures and side populations were sorted into Eppendorf tubes with 200,000 carrier cells. The carrier cells were pre-sorted FITC-CD19[+] B cells from spleen. The mixed cells were stained for 45 min with 20 μg/mL Hoechst 33342 and 100 μM Verapamil (Sigma) in DMEM+ media. Pyronin Y (Sigma-Aldrich) was added at 1 μg/mL and the incubation continued for 15 min. The cells were then washed, resuspended in PBS with 2% fetal bovine serum and immediately analyzed on a BD LSR Fortessa cell analyzer. The flow cytometry data was analyzed with FlowJo (v10.6.1).

**Serial syngeneic bone marrow transplantation**. For serial BMT, recipient BALB/c mice were lethally irradiated by 900 rad which was split into 2 doses (2 × 450 rad) and separated by 4 h. Four million total bone marrow cells from wild type mice or Sirt1[−/−] mice were transplanted into recipient mice by tail vein injection. To track the donor-derived cells, male BALB/c cells were used as donor cells for female recipients and male specific SRY genotyping analysis was performed to track them in bone marrow reconstitution. Knockout cells were further confirmed by SIRT1 genotyping. Mouse health was routinely monitored, and mice were euthanized according to the American Veterinary Medical Association Guidelines.

**DNA, RNA, and protein analyses**. For SRY genotyping, genomic DNA (gDNA) was extracted from blood mononuclear cells using DNeasy blood & tissue kit (QIAGEN). For V(D)J recombination assay, gDNA was extracted from sorted populations as indicated. About 100 ng gDNA per reaction was used for PCR. GAPDH or β-actin was used as housekeeping gene controls. All the primers were listed in the Supplementary Table 1. For routine RNA analysis, we extracted RNA using Directzol RNA Microprep Kit (Zymo Research) (with in-column DNase treatment), synthesized the first strand DNA with Superscript III kit (Invitrogen), and analyzed gene expression using SYBR Green qPCR SuperMix kit (KAPA biosystems). Sirt1 RT-PCR was performed with the forward primer on the intact exon 4 and the reverse primer on the exon 5 that was deleted in the KO mice. For Western blots, the harvested cells were lysed in RIPA/IP buffer (ThermoFisher Scientific) and protein extraction was quantified with a BCA protein assay kit (ThermoFisher Scientific). The Xcell-surelock mini gel System (ThermoFisher Scientific) was used for gel electrophoresis and protein transfer. The rabbit anti-mouse Sirt1 (Millipore, cat #07-131), rabbit anti-PHF20 (Cell signaling, cat #3934), and goat anti-mouse Trim26 antibody (Santa Cruz, sc-79774) were used.

**Gene knockdown using lentiviral vectors**. We made VSV-G (G protein of vesicular stomatitis virus) pseudotyped lentivirus using a four-plasmid transfection system as described previously[22]. High titer lentiviral stocks, typically $1–3 \times 10^7$ infectious units/ml, were used for transduction. Unless specified, a multiplicity of infection (MOI) around 5 was typically used for infection so that nearly complete transduction was achieved.

**SIRT1 knockout by CRISPR/Cas9 system in human primary MPAL and B-ALL cells**. Ficoll-enriched human B-ALL blasts were cultured in X-Vivo medium (Lonza) supplemented with 10% fetal bovine serum, 1% L-glutamine, 25 ng/mL hTPO, 50 ng/mL hSCF, 50 ng/mL hFlt3, and 20 ng/mL hIL3. Ficoll-enriched human primary B/M MPAL blasts were cultured in StemSpan SFEM II medium with 1× CD34+ Expansion Supplement (StemCell Technologies). Half medium change was performed every 2 days.

The sgRNAs targeting SIRT1 exon 2 was designed as follows: SIRT1+67887427 UUUGCAGAUAACCUUCUGUU; SIRT1−67887468 AUCCUCCUCAUCACUU UCAC. The sgRNAs with the scaffold sequence were synthesized by Synthego Inc. CRISPR/Cas9 ribonucleoprotein (RNP) system was used for gene editing. Briefly, 180 pmol of each sgRNA was incubated with 60 pmol Truecut Cas9 protein v2 (ThermoFisher Scientific) for 15 min at room temperature to form RNP complex. The RNP solution was diluted into 50 µL with electroporation buffer (Lonza) and kept on ice until use. Five million MPAL or B-ALL cells were collected by centrifugation at 300×$g$ for 10 min and followed by one time wash with phosphate-buffered saline. The cell pellet was then resuspended in Nucleofector P3 electroporation buffer (Lonza) and then mixed with RNP solution. The volume of each reaction is 100 µL and electroporation was carried out using a Lonza Nucleofector 4D system. For controls, the same procedures were performed on the cells without sgRNAs. After electroporation, the cells were diluted with their growth factor-supplemented media, seeded in 96-well plates at $1 \times 10^5$ cells/well and cultured as described above. On day 3 and day 6 after electroporation, the cells were analyzed. Apoptosis examination was performed by AnnexinV/DAPI staining and flow cytometry analysis. mRNA was extracted from the cells using Direct-Zol RNA miniprep kit (Zymo Research) for examining the expression levels of SIRT1 and B-cell differentiation genes by RT-qPCR. To determine the SIRT1 gene editing efficiency, Tracking of indels by decomposition (TIDE) assay was used to determine the spectrum and frequency of targeted mutations[80]. Briefly, genomic DNA was isolated and a 327 bp targeted fragment around the editing site was PCR amplified by the following primer set: forward primer 5′-GAAACTGAGTGAGC AGAATTCCTA and reverse primer: 5′-CTCTGAGCCATACCTATCCGTG. PCR products from wild type (controls) or gene edited samples were sequenced by Sanger sequencing. The ICE (Inference of CRISPR Edits) algorithm[81] was used to analyze the Sanger data, which is available online ice.synthego.com.

**Microarray gene expression analysis and bioinformatics**. For gene array analysis, RNA was extracted from flow cytometry sorted hematopoietic stem cells using PicoPure RNA isolation kit (Applied Biosystems) (with in-column DNase treatment). 500 pg RNA per sample was amplified using Ovation Pico WTA system V2 (NuGEN) and used for microarray analysis with Mouse Genome 430 2.0 gene chips (Affymetrix). Microarray data analysis was performed using Partek® Genomics Suite® (Partek Inc. release 6.12.0530). Microarray gene expression data was normalized using Robust Multi-array Average (RMA)[82] and summarized using the intensities of probes into gene-level expression. Differentially expressed genes were identified using one-way ANOVA with linear contrast between old WT or KO vs. young WT HSCs and old KO vs. old WT HSCs. Genes with statistically significant differential expression were selected by using |fold-change | > 1.5 at the level of mRNA expression and unadjusted $P$-value < 0.05 cutoff to generate heatmaps and Venn diagrams. The normalized signals for all samples were further analyzed using gene set enrichment analysis software (GSEA, v2.07, permute by gene set)[83,84] and a modified version of the c2 Canonical Pathways from the Molecular Signature Database (c2.cp MSigDB; v3.0)[85]. Ten extra gene sets (detailed in Supplementary Data 6) were extracted and/or modified from the published data[86–90] and added to c2.cp MSigDB for the GSEA analysis. The method for ranking genes was the default "Signal2Noise," and the scoring scheme was also the default "weighted". The resources of bioinformatic software packages are listed in "Supplementary Table 1".

In order to identify SIRT1-independent gene changes during aging, we looked at the overlap of probesets from (i) YWT vs. OWT (Old Wild-Type) and (ii) YWT vs. OKO (Old Knock-Out). We required changes to occur in the same direction. Additional overlap with the genes changed in OKO vs. OWT only resulted in a small difference in the above overlapping probeset lists, confirming that the genes identified were largely Sirt1-independent. The annotated Sirt1-independent overlapping probesets were provided in Supplementary Data 3.

We used three strategies for calculating pathway enrichment (Supplementary Data 4): (i) we used custom R code to calculate enrichment using a Fisher's Exact test for the 10 custom gene tests (with an FDR estimated using the method of Benjamini and Hochberg[91]; (ii) we calculated enrichment using Enrichr[92] (downloaded 9/23/2021) and downloaded the "MSigDB Hallmark 2020" enrichment results; (iii) we calculated enrichment using Ingenuity Pathway Analysis (Ingenuity® Systems, www.ingenuity.com analysis run on 9/23/2021, for each direction separately) and downloaded the IPA canonical pathways.

For IPA, probesets were used as the identifier, and the array design set to "Mouse Genome 430 2.0 Array" (in order to assist with the background correction). Mapped gene symbols were copied and pasted into the web interface for Enrichr, and manually removed any probes mapped to multiple genes (and no action taken to attempt a background correction). For the custom R code for the 10 custom gene sets, the Fisher's Exact test was used to compare the fraction of differentially expressed genes overlapping a gene set to the fraction of genes in the gene set overlapping the uniquely defined gene symbols on the array (after converting the mouse gene symbols to upper-case, to compare to the human gene lists). The Fisher's Exact test alternative was set to "greater" to only calculate significance for over-enrichment. Probesets mapped to multiple genes were excluded from analysis using the custom R code.

The heatmap for the Venn overlapping genes was created using the 'heatmap.2' function from the 'gplots' package (version 3.1.0) in R (version 3.3). Pearson dissimilarity was used as the distance metric, for RMA-normalized expression exported from Partek Genomics Suite. Venn diagrams were created using the 'Vennerable' package (version 3.0) in R (version 3.3), with 1 manual change made in the PDF version of the Venn diagram for a duplicated number (using Inkscape, version 0.91).

**Statistics and reproducibility**. For animal studies, Kaplan–Meier survival analysis was performed and statistical significance was calculated using log-rank test. For other data analysis, two-tailed $t$-test or Fisher's Exact test was performed in all cases and $P < 0.05$ is considered statically significant. The exact sample size for each experimental group/condition was shown in the data dot plots and measurements were taken from distinct biological replicates. Error bars were shown with standard deviation.

**Reporting summary**. Further information on research design is available in the Nature Research Reporting Summary linked to this article.

## Data availability

The raw microarray data reported in this paper have been deposited in GEO with the accession code GSE169387 (ref. [93]). GSEA reports of microarray data and Venn diagram analysis results were provided as Supplementary Data 1–6. The uncropped gel images were provided in Supplementary Figs. 9–12. Any remaining information can be obtained from the corresponding author upon reasonable request.

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

## Acknowledgements

This study is supported by the following grants: NIH R01 CA143421 and UH2/3 CA213385 sponsored by National Cancer Institute and National Institute of Aging, Norton Basic Research Fund, Tim Nesvig Lymphoma Research Fund, City of Hope Excellence Award and CUBRI award to W.Y.C. M.J.Y. is partially supported by an IRG from the University of Texas MD Anderson Cancer Center. The core facilities used in the study including the Animal Resources Center, Flow Cytometry Core and Integrative Genomics Core are supported by the National Cancer Institute under the award P30CA33572. The content is solely the responsibility of the authors and do not necessarily represent the official views of the NIH.

## Author contributions

Conceptualization, methodology and investigation: Z.W., W.Y.C. Data acquisition, analysis and curation: Z.W., C.Z., C.D.W., L.Z., Y.C.Y., C.G., C.W., J.W., X. Wu, W.Y.C. Providing key resources and service: M.J.Y., Y.L.Y., S.V., R.E., X. Wa, C.B., S.F. Writing: W.Y.C., Z.W. Funding acquisition: W.Y.C.

## Competing interests

W.Y.C. and Z.W. are listed on patent(s) related to this work. All other authors have no competing interest related to this work.
