## [Peer Review File · Communications Biology]

Reviewers' comments:

Reviewer #1 (Remarks to the Author):

This manuscript proposes a unique role of SIRT1 in aging HSCs which might be contrary to our general understanding. The work is interesting and may provide fresh ideas for the clinical treatment of MPAL.

1) In the discussion part, the author analyzes the probably reasons which cause the differences between their work and that of Singh et al. Additionally, there is another work previously published about SIRT1-deleted hematopoietic stem and progenitor cells (PMID: 25068121), the author should also take this work into consideration and improve the discussion part.

2) In line 351, the unpublished data should be presented at supplementary data.

Reviewer #2 (Remarks to the Author):

In the present work, Wang et al focus on the study of Sirt1 in aging and MPAL. Using a Sirt1 knockout mouse model they show how this factor promotes HSC aging and age-dependent B-cell differentiation block and development of B/M MPAL. Furthermore, they show how this factor is overexpressed in primary human MPAL and its knockdown inhibits growth and survival. Finally, they perform transcriptomic analyses to identify key genes and pathways that may be involved in MPAL development. Although the experiments are well thought and executed, there are some questions that arise when reading the manuscript:

Major comments:

1. The authors show how knockdown of SIRT1 in human B/M MPAL and B-ALL cell lines alters the expression of TCF3, PAX5 and EBF1. Are these alterations also found when this factor is knockdown in human primary samples?
2. It is stated that SIRT1 knockdown in cell lines inhibited the growth, induced the apoptosis, and sensitized cells to oxidative stress and chemotherapeutic agents. Although the apoptosis results are shown, the remaining are not. These results are very relevant and they should be shown either in the main or the supplemental figures.
3. Gene expression profiling is performed in OWT, YWT, and OKO HSCs. Although the level of expression of most genes in OKO resembles those of YWT cells, there is a group of genes (Figure 6A), overexpressed in OWT and OKO cells as compared to YWT cells, and that therefore represent genes upregulated in aging independently of Sirt1 expression. Could you describe which are those genes or in which processes/pathways are they involved in?

Minor comments:

1. Please check for typos throughout the manuscript.
2. In figure 6A, the legends are hard to read, please change them.
3. The numbers in figure 7F are partially overlapping, please correct them.

Reviewer #3 (Remarks to the Author):

In this study, Wang Z, et al. investigated the role of SIRT1 in aging HSC. They show that knocking out SIRT1 can 'delay' the aging of the HSC cells. For this HSC cells of old mice, both wildtype and with Sirt1 KO, are compared with one another and with control HSC cells from young mice. Furthermore, serial BMT are performed to further increase the age of the HSCs. Moreover, they describe this gene as a possible target for therapy in MPAL. To study the function of SIRT1, microarray analyses were performed.

Overall, this study is robust and nicely done. Although the part about SIRT1 in MPAL is not

extremely convincing, the role and function of SIRT1 in the aging of HSCs is well described and profoundly elaborated and this research is certainly an added value for the community.

Major comments:

1. I find it remarkable that 90% of the mice develop MPAL after the secondary/tertiary BMT with OWT. Although these cells are older, they still should be healthy HSCs. Why do these cells give rise to leukemia, just by the mutations introduced by aging? I would expect some of them indeed developing a hematological defect, but 90% seems high. How many mice were transplanted? Is this statistically significant? Please elaborate more.
2. Now the role of SIRT1 in MPAL is only addressed in these BMTs and a number of human cell lines. Would it be possible to have a PDX model with this disease? This is a suggestion, but it would be nice to have a PDX model and show that KO of SIRT1 impacts MPAL.
3. SIRT1 is proposed as a possible target for therapy of MPAL? Do you know of any inhibitors of this gene? If SIRT1 is involved in the early phase of HSC development, is it still possible to use this targeted therapy, in other words, will knocking out SIRT1 later on still reverse the effects? Please elaborate.

Minor comments:

1. The authors explain that *Sirt1*^{-/-} KO mice might show developmental issues. How frequently does this occur? Could you give a percentage of mice that survive this *Sirt1*^{-/-} KO?
2. To show the effect of SIRT1 on acetylation, only the marker H4K16ac is used. Can this be confirmed with other acetylation marks such as H3K27ac (active promoters/enhancers)?
3. For the GSEA analyses:
 - a. What genesets are exactly used? It is explained that 10 genesets were added, but what are they added to? To the C2 collection of MSigDB?
 - b. Are these GSEAs performed on ranked lists? What is used as a ranking score? Please elaborate.
 - c. Please add full tables of all analyses. Some analyses are only shown in plots (even without NES scores and FDR scores, Suppl. Figure 5). It would be more complete to also add full tables of these analyses.
4. The figures should be annotated in more detail. Some abbreviations are not explained (not in the figure, not in the legend, not in the text) and the figures are not always correctly annotated. Here are some of the things I noticed:
 - a. Figure 1A: WBC, NE, RBC, NY should be explained.
 - b. Figure 1B: what is measured here exactly?
 - c. Figure 1B: the abbreviations in the legend should be explained
 - d. Figure 2D: it should be explained what marker CD150 is? What does this mean?
 - e. Figure 4D: further annotate what is shown here? What do the arrows point to?
 - f. Figure 6B-C-D: what measure is used to rank the genes in this GSEA? What score is used to determine the up and downregulated genes in the comparisons (OKO vs OWT, OWT vs YWT)
 - g. etc.

Response to Reviewers

We thank three reviewers for their thorough review and constructive suggestions. We have performed additional benchworks and bioinformatic studies to address the concerns. The main changes were highlighted in the revised text. The point-to-point responses to comments are as follows.

Reviewer #1:

1) In the discussion part, the author analyzes the probably reasons which cause the differences between their work and that of Singh et al. Additionally, there is another work previously published about SIRT1-deleted hematopoietic stem and progenitor cells (PMID: 25068121), the author should also take this work into consideration and improve the discussion part.

We added discussion of Rimmele et al's study. For the comparison, we also added T cell change into Figure 1B, which was missing in the original submission. Overall, their phenotypes are mostly different from ours, but the influence of tamoxifen-induced Cre expression in their study could not be excluded. Besides, their study was conducted in young mice that had not undergone physiological aging process, and may not reflect true aging phenotypes. Further, their phenotypes are also not consistent with Sirt1 conditional knockout mice made with Mx-Cre. The revised discussion and more details were on p17.

2) In line 351, the unpublished data should be presented at supplementary data.

We added the data as new Supplemental Figure 2 and added a sentence on p6.

Reviewer #2:

Major comments:

1. The authors show how knockdown of SIRT1 in human B/M MPAL and B-ALL cell lines alters the expression of TCF3, PAX5 and EBF1. Are these alterations also found when this factor is knockdown in human primary samples?

We carried out CRISPR knockout of SIRT1 in human primary MPAL and B-ALL cells. We found similar changes of B-cell differentiation genes as shown in new Fig. 6A,B and the revised text on p11.

2. It is stated that SIRT1 knockdown in cell lines inhibited the growth, induced the apoptosis, and sensitized cells to oxidative stress and chemotherapeutic agents. Although the apoptosis results are shown, the remaining are not. These results are very relevant and they should be shown either in the main or the supplemental figures.

We added growth curves on Fig. 5E and moved supplemental apoptosis data to Fig. 5F, and revised the text on p11.

3. Gene expression profiling is performed in OWT, YWT, and OKO HSCs. Although the level of expression of most genes in OKO resembles those of YWT cells, there is a group of genes (Figure 6A), overexpressed in OWT and OKO cells as compared to YWT cells, and that therefore represent genes upregulated in aging independently of Sirt1 expression. Could you describe which are those genes or in which processes/pathways are they involved in?

We performed new Venn diagram analysis to identify those genes exhibiting age-related changes but independent of Sirt1 as shown in new Supplementary Fig. 8. We identified 156 genes in this group and they were listed in the new Supplementary Table 3. These genes affected pathways including TNF α signaling via NF-kB, estrogen responses, hypoxia, E2F targets, G2-M checkpoint, etc, as detailed in new Supplementary Table 4. The text was added on p13 and in the Methods section.

Minor comments:

1. *Please check for typos throughout the manuscript.*

We fixed numerous typos throughout the manuscript.

2. *In figure 6A, the legends are hard to read, please change them.*

We regenerated Fig. 6A (now Fig. 7A) with higher resolution and it is more legible now.

3. *The numbers in figure 7F are partially overlapping, please correct them.*

The overlapping was corrected.

Reviewer #3:

Major comments:

1. *I find it remarkable that 90% of the mice develop MPAL after the secondary/tertiary BMT with OWT. Although these cells are older, they still should be healthy HSCs. Why do these cells give rise to leukemia, just by the mutations introduced by aging? I would expect some of them indeed developing a hematological defect, but 90% seems high. How many mice were transplanted? Is this statistically significant? Please elaborate more.*

We added more details of our MPAL mice on p7. The acute leukemia was found in 18 out of 20 (i.e. 90%) mice receiving OWT bone marrow cells in three independently completed cohorts in which all these mice died of diseases (excluding those found dead and decomposed), which contrasted with only 2 cases of acute leukemia in 29 mice receiving OKO cells in the cohorts ($P < 0.0001$). Because BMT was conducted with total donor bone marrow cells, we agree with the reviewer that some healthy old HSCs were still there, which helped most recipients survive first couples of months after BMT as shown in Fig. 3C-E. But later, the leukemia stem cell clones emerged and developed into MPAL. We agree that it is important to learn how leukemia stem cell clones are actually emerged. Although we don't know the exact mechanisms at present, our array data pointed to alterations of numerous gene and pathway changes in aging HSCs that may facilitate oncogenic transformation and the emergence of leukemia stem cell clones, which was inhibited by Sirt1 KO.

2. *Now the role of SIRT1 in MPAL is only addressed in these BMTs and a number of human cell lines. Would it be possible to have a PDX model with this disease? This is a suggestion, but it would be nice to have a PDX model and show that KO of SIRT1 impacts MPAL.*

We agree with the reviewer that a PDX model of MPAL will be very useful for the study. However, human MPAL samples are rare and extremely difficult to collect. At present, we have not established a PDX model. With that said, we have now successfully grown human primary MPAL cells in growth factor-supplemented medium. As a result, we performed SIRT1 KO by CRISPR in human primary

MPAL cells (new Fig. 6). As mentioned above in response to Reviewer 2, our results showed that SIRT1 KO increased expression of B-cell differentiation genes and induced apoptosis in human primary MPAL cells similar to that in human cell lines.

3. SIRT1 is proposed as a possible target for therapy of MPAL? Do you know of any inhibitors of this gene? If SIRT1 is involved in the early phase of HSC development, is it still possible to use this targeted therapy, in other words, will knocking out SIRT1 later on still reverse the effects? Please elaborate.

As shown in new Fig. 6 and mentioned above, SIRT1 KO in human primary MPAL increased expression of B-cell differentiation genes and induced apoptosis. Our results provide a proof-of-principle of SIRT1 KO effect on human MPAL, despite our CRISPR KO efficiency in MPAL cells was not high enough and there was high background of cell death. We also tested SIRT1 inhibitor tenovin-6 and showed that it induced significant apoptosis of human primary MPAL cells (new Fig. 6D). Our new results add further support for targeting SIRT1 for human MPAL. Future studies will examine the effects of SIRT1 inhibition in combination with other drugs to amplify the effect of killing MPAL cells.

Minor comments:

1. The authors explain that Sirt1^{-/-} KO mice might show developmental issues. How frequently does this occur? Could you give a percentage of mice that survive this Sirt1^{-/-} KO?

All Sirt1^{-/-} mice in the BALB/c strain suffered some early development-related health issues including lower bodyweight and a microphthalmia-like condition for the entire life. About half of Sirt1^{-/-} mice lived through adulthood in this strain. This was added on p5.

2. To show the effect of SIRT1 on acetylation, only the marker H4K16ac is used. Can this be confirmed with other acetylation marks such as H3K27ac (active promoters/enhancers)?

We analyzed H4K16ac because it is the most validated SIRT1 histone substrate in vitro and in vivo (Imai, S.I et al. 2000, Nature; Vaquero A, et al .2004 Mol Cell and 2007 Nature). Although the changes in acetylation of H3 K27 and possibly some other histone marks may occur, they may be indirect results and do not offer direct insight of the SIRT1 KO.

3. For the GSEA analyses:

a. What genesets are exactly used? It is explained that 10 genesets were added, but what are they added to? To the C2 collection of MSigDB?

Ten extra genesets were detailed in Supplementary Table 6 and they were added to c2.cp MSigDB for the GSEA analysis. The Methods section was revised and clarified on p26.

b. Are these GSEAs performed on ranked lists? What is used as a ranking score? Please elaborate.

GSEA is being provided RMA (Robust Multi-array Average) normalized Mouse 430 2.0 probe expression. The method for ranking genes was the default “Signal2Noise,” and the scoring scheme was also the default “weighted”. This was added in the Methods section on p26.

c. Please add full tables of all analyses. Some analyses are only shown in plots (even without NES scores and FDR scores, Suppl. Figure 5). It would be more complete to also add full tables of these analyses.

The full tables were provided and we added sentences to clarify that in the main text on p12 as well as the figure legends.

4. The figures should be annotated in more detail. Some abbreviations are not explained (not in the figure, not in the legend, not in the text) and the figures are not always correctly annotated. Here are some of the things I noticed:

a. Figure 1A: WBC, NE, RBC, NY should be explained.

The annotations for these abbreviations were added in the Figure 1A legend.

b. Figure 1B: what is measured here exactly?

Figure 1B legend was clarified for flow cytometry analysis of bone marrow cell lineages.

c. Figure 1B: the abbreviations in the legend should be explained.

We believe this comment was for Figure 1F of CFU colonies. We added annotations for all abbreviations.

d. Figure 2D: it should be explained what marker CD150 is? What does this mean?

We added a brief explanation of CD150 marker for HSCs and its role in aging HSCs in Figure 2D legend.

e. Figure 4D: further annotate what is shown here? What do the arrows point to?

More detailed annotation of Figure 4D was provided.

f. Figure 6B-C-D: what measure is used to rank the genes in this GSEA? What score is used to determine the up and downregulated genes in the comparisons (OKO vs OWT, OWT vs YWT)

As mentioned above, the method for ranking genes was the default “Signal2Noise,” and the scoring scheme was also the default “weighted”. GSEA plots were ranked by NES scores and the full sets of results were provided in Supplementary Tables 1 and 2. The Figure legend and the Methods section were revised to make this clearer.

In addition to the above changes, we added more details about animal housing conditions in the Materials and Methods. We also added Drs. Chunxiao Zhang and Christine Brown for their help in collecting revision data.

REVIEWERS' COMMENTS:

Reviewer #2 (Remarks to the Author):

This reviewer has been satisfied with the authors corrections and has no further questions. I am therefore happy to accept the paper as it is.

Reviewer #3 (Remarks to the Author):

In this manuscript, the role of SIRT1 in aging HSC is investigated. This study is well performed and documented and certainly an added value for the community.

The comments of the the reviewers were nicely addressed, which in my opinion further increases the quality of this study.

I have no further comments.